Journal of Data-centric Machine Learning Research (2024)          Submitted 5/24; Revised 8/24; Published 9/24

# On Catastrophic Inheritance of Large Foundation Models

**Hao Chen**                                                     HAOC3@ANDREW.CMU.EDU
*Carnegie Mellon University*

**Bhiksha Raj**                                                  BHIKSHAR@ANDREW.CMU.EDU
*Carnegie Mellon University*

**Xing Xie**                                                     XINGX@MICROSOFT.COM
*Microsoft Research*

**Jindong Wang**                                                 JINDONG.WANG@MICROSOFT.COM
*Microsoft Research, William & Mary*

**Reviewed on OpenReview:** *https://openreview.net/forum?id=fONi0rnjLE&referrer= %5BAuthor%20Console%5D*

**Editor:** Andreas Kirsch

## Abstract

Large foundation models (LFMs) are claiming incredible performances. Yet great concerns have been raised about their mythic and uninterpreted potentials not only in machine learning, but also in various other disciplines. In this position paper, we propose to identify a neglected issue deeply rooted in LFMs: *Catastrophic Inheritance*, describing the weaknesses and limitations inherited from biased large-scale pre-training data to behaviors of LFMs on the downstream tasks, including samples that are corrupted, long-tailed, noisy, out-of-distributed, to name a few. Such inheritance can potentially cause catastrophes to downstream applications, such as bias, lack of generalization, deteriorated performance, security vulnerability, privacy leakage, and value misalignment. We discuss the challenges behind this issue and propose "UIM", a framework to *Understand* the catastrophic inheritance of LFMs from both pre-training and downstream adaptation, *Interpret* the implications of catastrophic inheritance on downstream tasks, and how to *Mitigate* it. UIM aims to unite both the machine learning and social sciences communities for more responsible and promising AI development and deployment.

**Keywords:** Catastrophic Inheritance, Pre-training Data, Foundation Models

## 1 Introduction

In the rapidly evolving landscape of machine learning, large foundation models (LFMs), such as CLIP (Radford et al., 2021; Cherti et al., 2023), GPT (Radford et al., 2019; Brown et al., 2020; OpenAI, 2023), PaLM-2 (Anil et al., 2023), LLaMA (Touvron et al., 2023a,b), Stable Diffusion (Rombach et al., 2022), Gemini (Google, 2023), Time-LLM (Jin et al., 2023), etc, have emerged as a cornerstone (Bommasani et al., 2021) for numerous real-world

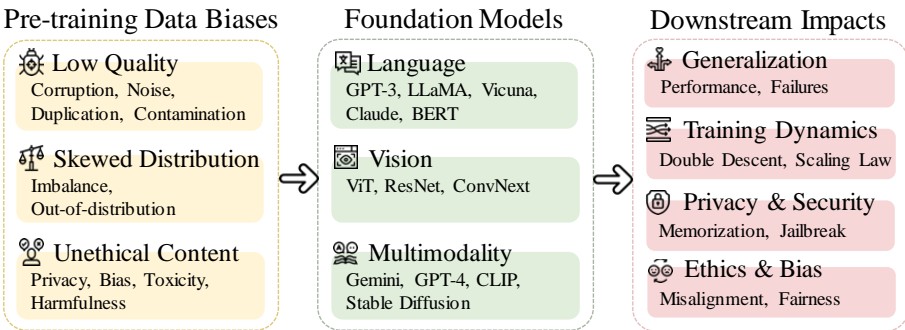

Figure 1: Illustration of catastrophic inheritance. Large foundation models pre-trained on biased datasets may cause significantly malicious consequence to various downstream tasks (rf. Table 1).

tasks. Characterized by their large parameter sizes and extensive training on large-scale data (Sevilla et al., 2022), LFMs have demonstrated remarkable abilities such as zero-shot learning (Radford et al., 2021; Gruver et al., 2024) and in-context learning (Radford et al., 2019; Brown et al., 2020; Gao et al., 2020b; Kaplan et al., 2020; Zhao et al., 2021; Wei et al., 2022; Olsson et al., 2022; Rasul et al., 2023), and impressive transfer performance across various tasks (Zhuang et al., 2020; He et al., 2021; Nakkiran et al., 2019; Kaplan et al., 2020). As LFMs claim promising performances in almost every discipline from computer science, natural science, to social science, it is urgent yet challenging to fully evaluate and understand their capabilities, limitations, and failures.

This paper proposes a phenomenon and novel research direction – *Catastrophic Inheritance*, describing that LFMs pre-trained on increasingly large-scale but biased datasets can cause potentially significant and catastrophic consequences to downstream tasks (Caballero et al., 2022; Schaeffer et al., 2023). As evidenced in Table 1, various user applications that rely on LFMs are affected by potentially biased pre-training datasets from multiple aspects, including ethics (Forbes, 2023; Thiel, 2023), security (Wang et al., 2018; Zhang et al., 2022), generalization (Chen et al., 2024b), language understanding (Jin et al., 2024), and culture bias (Boston.com, 2023), to name a few. For example, LAION-5B (Schuhmann et al., 2022), the popular pre-training dataset for Stable Diffusion and many other LFMs, is reported to contain harmful content (Birhane et al., 2023), such as child sexual abuse material (Forbes, 2023), which was then inherited to Stable Diffusion models to generate similar harmful contents. Existing research also shows that biases in the pre-training data inevitably perturb and maliciously affect the generalization and behaviors of LFMs (Dodge et al., 2021; Chen et al., 2024b; Dong et al., 2023; Longpre et al., 2023). Perhaps more alarmingly, the detrimental effects of biases might be concealed superficially after fine-tuning on specific downstream tasks (Jain et al., 2023b; Qi et al., 2023), which may consequently raise safety and security concerns in the deployment (Carlini et al., 2023a; Gu et al., 2023; Mallen et al., 2022). In essence, despite the difference in architectures and proxy pre-training tasks of LFMs (Vaswani et al., 2017; Ronneberger et al., 2015), the myth of their training behaviors and capabilities (Nakkiran et al., 2019; Power et al., 2022; Kaplan et al., 2020) largely inherit from the opaque and large-scale pre-training datasets (Entezari et al., 2023; Elazar et al., 2023).

With the rapid evolution of LFMs, scaling the dataset from web-collected contents (Raffel et al., 2020; Gao et al., 2020a; Schuhmann et al., 2022; Byeon et al., 2022; Computer, 2023; Penedo et al., 2023; Soldaini et al., 2023) becomes a convention to improve model generalization, which avoids heavy human efforts of curation and annotation (Birhane et al., 2023). However, does scaling really beat (the effect of) biases? The increasingly scaled Internet data also inevitably contains more *imbalanced* (Reed, 2001; Zhu et al., 2023a; Parashar et al., 2024), *duplicated* (Lee et al., 2022; Hernandez et al., 2022; Tirumala et al., 2023; Elazar et al., 2023; Xu et al., 2023b), *corrupted* (Luccioni and Viviano, 2021; Birhane et al., 2023; Carlini et al., 2023a; Yang et al., 2023b), *contaminated* (Marone and Van Durme, 2023; Wei et al., 2023b; Oren et al., 2023; Deng et al., 2023; Jiang et al., 2024), *noisy* (Kolesnikov et al., 2020; Schuhmann et al., 2022; Chen et al., 2024b), and even *unethical* and *biased* (Forbes, 2023; Thiel, 2023; Jin et al., 2024) samples. Even more recently, synthetic data has been widely involved in the pre-training of large language models (Gunasekar et al., 2023) and time-series foundation models (Dooley et al., 2024). While showing promising downstream performance, the limitation of synthetic data should also be considered (Alemohammad et al., 2023). Furthermore, some of the advanced and proprietary LFMs such as GPT-4 (OpenAI, 2023) and Gemini (Google, 2023) do not open-source their training data. The huge volume, high complexity, and black-box nature of the pre-training data make it economically expensive and technically impossible to detect and remove all the biased samples, which thus maliciously affect the LFMs' behavior and generalization.

In this paper, we propose *UIM*, a general research framework to understand, interpret, and mitigate the catastrophic inheritance of LFMs to the downstream tasks. Despite the prosperous development and research to improve the generalization of LFMs, addressing catastrophic inheritance has received limited attention and presents a few unsolved challenges. First, it remains unclear how the pre-training data biases will directly affect the generalization properties and the training dynamics (Nakkiran et al., 2019; Kaplan et al., 2020; Power et al., 2022) of these models at the pre-training stage, which may inherit to the subsequent tasks (Caballero et al., 2022; Dar et al., 2021). Second, it remains unknown how to interpret the effects of biased pre-training data on downstream tasks and the fundamental reasons for these effects from LFMs (Bender et al., 2021; Jain et al., 2023b; Chen et al., 2024b). The lack of comprehensive evaluation and proper metrics of LFMs beyond the performance of downstream tasks is one of the most essential reasons impeding our understanding and interpretation of catastrophic inheritance (Sun et al., 2023; Lee et al., 2023; Schaeffer et al., 2023; Tong et al., 2024). Third, due to the black-box and complex nature of LFMs and pre-training datasets, it becomes notoriously difficult to mitigate the malicious effects of pre-training biases on downstream tasks (Oren et al., 2023; Chen and Yang, 2023; Chen et al., 2024b; Zhang et al., 2024b), without re-train the model from scratch. To overcome these challenges, UIM involves three aspects:

- **Understanding** the catastrophic inheritance from pre-training dynamics, generalization behaviors, scaling laws, effects on downstream tasks, with more comprehensive evaluation benchmarks and effective metrics of LFMs.

- **Interpreting** the fundamental sources in LFMs that lead to catastrophic inheritance on generalization to downstream tasks, both empirically and theoretically.

Table 1: Realistic examples of catastrophic inheritance from published papers or news.

| Example | Domain | Source |
| --- | --- | --- |
| Stable Diffusion models was trained on Laion-5B, which contains hundreds of harmful images of child sexual abuse material (CSAM). Then, the model was reported to memorize during training and generate CSAM at production. | Ethics and privacy | (Birhane et al., 2023; Forbes, 2023; Thiel, 2023) |
| At least 50% of poisoning, adversarial, and backdoor vulnerabilities will be inherited from pre-training data to fine-tuned models, which can be easily triggered at the deployment. Jailbreaks may also relate to pre-training biases. | Security | (Wang et al., 2018; Zhang et al., 2022; Carlini et al., 2023a; Zou et al., 2023) |
| An MIT student asked AI to make her headshot more 'professional.' It gave her lighter skin and blue eyes. Country bias also found in language models. | Bias | (Boston.com, 2023; Wang et al., 2023c) |
| Fine-tuning LLMs on only 10 adversarially designed or even benign samples leads to degradation of safety alignment, which costs less than $0.2 using API. | Misalignment | (Qi et al., 2023) |
| Noisy labels contained in pre-trained data always hurt downstream OOD performance; more than 10% noisy data will hurt in-domain performance. | Generalization | (Chen et al., 2024b) |
| Large language models like GPT-3.5 exhibited an accuracy reduction of 18.12% when answering non-English medical questions. Similar for coding tasks. | Model behaviors | (Jin et al., 2024; Zheng et al.) |
| Noise in the pre-training data strengthen the double descent phenomena, where the critical point of LFMs overfitting/memorizing data appears earlier. | Training dynamics | (Nakkiran et al., 2019) |

- **Mitigating** catastrophic inheritance on downstream tasks in (partially) black-box paradigms without re-training LFMs from scratch, access to full architecture/weights of LFMs, and access to large-scale pre-training datasets.

UIM stands out as a set of under-explored research directions that could trigger many new opportunities not only in connecting traditional machine learning efforts to LFMs but also in unprecedented interpretation of LFMs, including vision, language, and most importantly, social sciences. Since the impacts of LFMs lie not only in the algorithmic level, but in societal level that matters to everyone. The involvement of social sciences is indispensable to help researchers better evaluate the capabilities of models, measure societal impact, design human study, and delve into all aspects of society for risk management. We hope these research topics will facilitate a better understanding on the generalization of foundation models from both the stage of pre-training and downstream tasks transferring, which ultimately helps us curate more high-quality pre-training datasets and build more promising models. The remainder of the paper is organized as follows. In Section 2, we introduce the relevant background, formal definition of catastrophic inheritance, and preliminary studies in the relevant fields. We then identify the challenges of understanding and solving catastrophic inheritance in Section 3 and present our proposals for future research in each dimension with more details in Section 4. At the end, we conclude this position paper in Section 5.

## 2 Catastrophic Inheritance

In this section, we present a comprehensive review of literature related to catastrophic inheritance, defined as:

**Definition 1** *Catastrophic Inheritance (CI) refers to as the catastrophic and malicious impacts of adapting large foundation models $\mathcal{M}$ on downstream tasks with data $\mathcal{D}_{\mathrm{down}}$ and algorithm $\mathcal{A}_{down}$, which are learned and inherited from the large-scale but potentially biased pre-training data $\mathcal{D}_{\mathrm{up}}$ with the pre-training proxy algorithm $\mathcal{A}_{\mathrm{up}}$:*

$$\mathrm{CI} = g\left(\mathcal{D}_{\mathrm{down}}, f\left(\mathcal{D}_{\mathrm{up}}, \mathcal{M}, \mathcal{A}_{\mathrm{up}}\right), \mathcal{A}_{\mathrm{down}}\right), \tag{1}$$

*where $f$ corresponds to the pre-trained model that encompasses the change of models' behaviors, capacities, and generalization, and $g$ models the malicious impacts on downstream subjected to both pre-training and downstream.*

The catastrophic inheritance thus models a function of both downstream and pre-training model, dataset, and algorithm. We review the recent realistic examples of catastrophic inheritance (Section 2.1), the related works from the biases in pre-training data $\mathcal{D}_{\mathrm{up}}$ (Section 2.2), the potential impacts of such biases $g$ on downstream tasks (Section 2.3), and the rather underexplored mitigation strategies on the malicious effects of them, which reduces $g$, (Section 2.4), especially the black-box methods due to the limited access of LFMs.

### 2.1 Realistic Examples of Catastrophic Inheritance

Here, we present realistic examples that underscore the concept of catastrophic inheritance, as shown in Table 1.

Studies (Wang et al., 2018; Rezaei and Liu, 2020) have indicated that fine-tuned models can inherit issues from pre-trained models containing backdoor vulnerabilities. This risk is amplified in large-scale pre-training datasets, as Carlini et al. (2023a) demonstrated, which are prone to being poisoned, thus adversely affecting downstream tasks. Zhang et al. (2022) found a significant probability (approximately 50%) that downstream fine-tuned models inherit adversarial and backdoor vulnerabilities from their pre-trained counterparts. The massive capacity of LFMs often leads to the memorization of these harmful samples, which could manifest at deployment, thereby raising severe security, privacy, and bias concerns (Birhane et al., 2023; Qi et al., 2023).

Moreover, Chen et al. (2024b) have shown that the presence of noisy labels in pre-training data consistently undermines performance in out-of-distribution tasks. The noise inherent in pre-training data also affects the training dynamics of LFMs (Nakkiran et al., 2019). Jin et al. (2024) reported a notable decrease in accuracy (about 18.12%) by GPT-3.5 in responding to non-English medical inquiries. The corresponding findings were reported by Zheng et al., showing that language models pre-trained in English tend to outperform those trained in Chinese on Chinese-language tasks. However, comprehensive methods for understanding, interpreting, and mitigating the effects of catastrophic inheritance in LFMs remain largely undeveloped.

Table 2: Common pre-training data biases identified in previous literature.

| Bias Type | Biased Data & Def. | Malicious Effects | Source |
|---|---|---|---|
| Low Quality | Duplication: Exactly the same and semantically similar content/samples | Memorization, privacy risks | Elazar et al. (2023); Carlini et al. (2022); Hernandez et al. (2022) |
| Low Quality | Corruption/Noise: Unnatural and Unmatched inputs and supervision | Deteriorated generalization and performance on downstream | Elazar et al. (2023); Fan et al. (2023a); Kreutzer et al. (2022) |
| Low Quality | Contamination: Leakage of testing samples to training data | Broken and inaccurate evaluation | Roberts et al. (2023); Schaeffer (2023); Jiang et al. (2024) |
| Skewed Dist. | Imbalance: Concepts clusters form different and imbalanced proportion | Biased predictions from rare concepts with worse performance | Xu et al. (2023c); Zhu et al. (2023a); Parashar et al. (2024) |
| Unethical Content | Biases, toxicity, and harmfulness | Harmful generation | Zou et al. (2023); Sun et al. (2024) |

## 2.2 Biases in the Pre-training Data

We discuss the common biases in the pre-training data $\mathcal{D}_{\text{up}}$ identified in the previous literature, as shown in Table 2. We summarize three types of pre-training data biases: low quality, skewed distribution, and unethical content.

**Low quality**. Low-quality training samples can be prevalent in large-scale, web-crawled pre-training datasets, which includes and not limits to data duplication, corrupted, noisy, and contaminated data. These biases directly affect LFMs' behaviors and capabilities on various downstream tasks (Hall et al., 2022).

Repeated samples have been reported as a common occurrence in the pre-training data. Studies (Kandpal et al., 2022; Elazar et al., 2023) have revealed a significant percentage of duplicates in datasets such as RedPajama (Computer, 2023) and Laion-2B-en (Schuhmann et al., 2022). This repetition not only affects the efficiency of the learning process but also presents memorization issues that lead to privacy risks, as discussed by Carlini et al. (2022) and Lee et al. (2022). Hernandez et al. (2022) also studied the scaling laws and interpretability of training language models duplicated in a systematic manner, confirming that repeated data can negatively impact the learned structures crucial for generalization. Many recent research has focused on the de-duplication of the pre-training data with various techniques (Coupette et al., 2021; Abbas et al., 2023) and found improved generalization when training on filtered and deduplicated data (Penedo et al., 2023; Tirumala et al., 2023).

Corrupted and noisy samples and supervision are prevalent, encompassing broader issues than traditional noisy label learning (Natarajan et al., 2013), including unmatched pairs in multimodal datasets and low-quality elements in self-supervised pre-training. Kreutzer et al. (2022) highlighted the presence of such low-quality texts in web-scale datasets, especially in low-resource languages. Similarly, Gunasekar et al. (2023) and Zheng et al. observed performance disparities in language models trained on different language codes and sources. Jain et al. (2023a) supported the idea that structured data leads to better results in tasks

such as code generation. Recent trends include using synthetic data for pre-training, which, if of low quality, can also introduce the corruption to pre-training. Noise and corruption in the pre-training data can impose impacts on the behaviors of the models and generalization to downstream tasks in various dimensions (Longpre et al., 2023; Chen et al., 2024b).

Data contamination in LFMs, where training data overlaps with test data, has also been increasingly recognized as problematic. It challenges our understanding of LFMs' true capabilities (Dodge et al., 2021; Yang et al., 2023a; Roberts et al., 2023; Li, 2023b; Deng et al., 2023; Jiang et al., 2024). Studies Schaeffer (2023) showed that training on test data can disrupt expected scaling laws and induce grokking behaviors. Li and Flanigan (2023) found that LLMs perform differently depending on the date of creation of the test data. Recognizing the overlap of training and testing data, new metrics for detecting contamination have emerged, such as loss difference (Wei et al., 2023b), model-based (Yang et al., 2023a), perplexity (Li, 2023a), and black-box method (Oren et al., 2023) without access to pre-training data or model.

**Skewed Distribution**. The concepts/clusters/subsets in the web-collected pre-training data often exhibit long-tailed distributions (Reed, 2001) that are difficult to re-balance at scale, casting challenges to most of the self-supervised LFMs (Kandpal et al., 2023). The imbalance skews LFMs' capabilities towards more frequent concepts, as demonstrated by Zhu et al. (2023a). For instance, the CLIP (Radford et al., 2021) and MetaCLIP (Xu et al., 2023c) models show better generalization than those trained on LAION-400M (Schuhmann et al., 2021; Cherti et al., 2023), due to their "balanced" data curation strategy. Instead of naively scraping web data, CLIP and MetaCLIP collected at most 20K image-text pairs for each of 500K visual concepts. Despite efforts to data balancing, many visual concepts remain underrepresented, with fewer than 20K samples (Xu et al., 2023c). Consequently, applications reliant on pre-trained CLIP models, including vision-language chatbots (Liu et al., 2023a; OpenAI, 2023) and text-to-image generative models (Rombach et al., 2022), also fail to recognize or generate images featuring rare concepts (Parashar et al., 2024).

**Unethical Content**. The pre-training data for LFMs often contains content that is private, harmful, biased, or toxic, leading to significant risks in public safety, social security, and trust, particularly at the deployment of these models. Inherent biases, including gender (Kotek et al., 2023b), cultural (Tao et al., 2023), racial biases (Omiye et al., 2023), and stereotypes (Ma et al., 2023), are often reflected in these models, most likely inherited from the pre-training data. Unsafe and harmful content has been continuously reported (Jansen et al., 2022). This concern is highlighted in studies such as Dodge et al. (2021); Yao et al. (2023); Sun et al. (2024), and Kotek et al. (2023a), stressing the importance of careful data curation for model training. Addressing these issues not only improves the reliability of LFMs but also contributes to social science with more responsible AI.

**Pre-training data inspection tools**. Recognizing the various biases and issues in LFM pre-training data, a range of inspection tools and protocols have been developed. These include Data Portraits (Marone and Van Durme, 2023) for detecting test set leakage and model plagiarism, the Laion-2B retrieval engine (Schuhmann et al., 2022) for visualizing image-text pairs, the Text Characterization Toolkit (TCT) (Simig et al., 2022) for analyzing large dataset characteristics, searching tools (Piktus et al., 2023a,b) for qualitative analysis. The more recent Oasis (Zhou et al., 2023) offers a system for data quality assessment, and WIMBD (Elazar et al., 2023) enables fast data search and counting. The tools of more

functions documenting and understanding the pre-training data are crucial for documenting and understanding pre-training data, which is key to developing better, more effective, and well-regulated LFMs (Mitchell et al., 2022).

### 2.3 Potential Impacts to Downstream Tasks

In this section, we explore how biases in pre-training data potentially impact downstream tasks, i.e., $g$ of LFM. These biases are evidenced to significantly affect training dynamics, generalization, security, and lead to misalignment and fairness issues. Understanding and interpreting these impacts is crucial for enhancing LFMs' performance and reliability, especially for applications related to society science.

**Training Dynamics**. Biases in pre-training data can significantly influence the training dynamics of LFMs, thus affecting their performance on downstream tasks. The double descent phenomenon (Opper, 1995; Belkin et al., 2019; Nakkiran et al., 2019), where model performance initially decreases before improving with increasing model and dataset scale, is found to be exacerbated by data biases. The transfer of double descent behavior to downstream tasks also potentially indicates a direct inheritance of affected pre-training characteristics (Dar et al., 2021). Scaling laws (Kaplan et al., 2020), which relate loss to dataset scale, model size, and training time, are critical to predicting model behaviors and the downstream performance of larger models from smaller ones. However, broken scaling laws (Caballero et al., 2022), indicating deviations in these predictions, suggest that biases in pre-training data might disrupt the expected behaviors and affect downstream generalization and performance (Cherti et al., 2023). Grokking behavior (Power et al., 2022; Varma et al., 2023a), describing the sudden spike in generalization from random to perfect levels, often occurs beyond the point of overfitting. This behavior has been linked to a transition from memorization to generalization (Kumar et al., 2023; Davies et al., 2023; Varma et al., 2023b). The correlation between training dynamics and model structure, such as induction heads in Transformers (Olsson et al., 2022; Reddy, 2023), implies that biases in pre-training data could also influence critical model functions. Understanding how these biases affect the critical data size for such dynamic changes requires further investigation (Zhu et al., 2024).

**Generalization**. The connection between pre-training data biases and the generalization on downstream tasks is crutial in the LFM era. Previous study (Recht et al., 2019) revealed the significant influence of data collection biases, such as those in ImageNet, on transfer performance. Although data diversity improves robustness and generalization (Fang et al., 2022; Ramanujan et al., 2023; Entezari et al., 2023), especially in real-world datasets (Fang et al., 2023; Richards et al., 2023), it often comes at the cost of quality (Nguyen et al., 2022). Merely increasing the quantity cannot guarantee the diversity always. This trade-off is exemplified in the findings of Abnar et al. (2021) and Tu et al. (2023), showing how limited data diversity and inherent biases impact the reliability and robustness of the model. The balance between intra-class and inter-class diversity also remains a complex issue to solve (Shirali and Hardt, 2023; Zhang et al., 2023a). Data pruning methods (Sorscher et al., 2022; Marion et al., 2023; Abbas et al., 2024; Fu et al., 2024) have recently been widely studied to purify the quality, improving the generalization of LLMs.

Recent studies focus more on the specific impacts of bias in pre-training data on downstream tasks, revealing nuanced effects on in-distribution and out-of-distribution performance

that may present negative correlation (Wenzel et al., 2022; Shi et al., 2023). Chen et al. (2024b) found that slight noise in pre-training data can benefit the ID performance, while always hurting the OOD performance. Hernandez et al. (2022) and Tong et al. (2024), studied data repetition and noisy image-text pairs in pre-training to increased memorization and general failures in LFMs, respectively. Yamada and Otani (2022) found that the robustified model in pre-training usually also presents robustness in downstream tasks. This evolving research area is critical to understanding catastrophic inheritance and improving the robustness and generalization of LFMs in real-world applications.

**Privacy and Security**. Pre-training data biases, especially those related to duplication and private information, can raise severe privacy and security issues of LFMs (Wei et al., 2023a; Bagdasaryan et al., 2023; Zou et al., 2023; Yao et al., 2023; Kumar et al., 2024; Li et al., 2024). LFMs can be elicited to verbally output private information that has been memorized at production (Carlini et al., 2023b; Nasr et al., 2023). The property of LFMs being universally attacked (Zou et al., 2023; Duan et al., 2023) and the jailbreak of LLMs (Huang et al., 2023; Chao et al., 2023; Wyllie et al., 2024) may also relate to the pre-training biases, but unidentified due to the lack of proper evaluation. These weaknesses of LFMs can usually not be found until they occur in practice, highlighting the necessity of more evaluation benchmarks from privacy and security.

**Ethics and Bias**. Biases in pre-training can also post vulnerabilities related to society science at the deployment of LFMs on downstream tasks, including misalignment (Wolf et al., 2023; Yang et al., 2023c), bias and fairness (Gallegos et al., 2023), and unethical content generation (Tokayev, 2023), affecting their reliability in critical applications like medical or financial systems. The misuse and unsafe deployment of LFMs (Mozes et al., 2023) also reflects the lack of comprehensive evaluation and proper metrics of them. Addressing the catastrophic inheritance of these biases is crucial for ensuring the reliability and fairness of LFMs.

### 2.4 Mitigation

Mitigating the impact of the biased pre-training data on LFMs, i,e, reducing $g$ without full access and control of $f(\mathcal{D}_{up}, \mathcal{M}, \mathcal{A}_{up})$, is a complex and challenging task. The straightforward approach of identifying and filtering biases is difficult due to the need to maintain data diversity and quantity (Simig et al., 2022; Touvron et al., 2023a). Re-training LFMs can effectively mitigate specific biases, but requires significant computational resources and may introduce new issues (Longpre et al., 2023). Alternative strategies include unlearning techniques (Bourtoule et al., 2021; Xu et al., 2023a), allowing models to forget harmful biases (Bourtoule et al., 2021; Jang et al., 2022; Wu et al., 2024), and black-box methods that mitigate biases without full access to the model and data (Chen et al., 2024b; Oren et al., 2023). These approaches aim to balance bias mitigation with the practicalities and limitations of LFM tuning.

### 3 Challenges of Catastrophic Inheritance

We introduced and explored the concept and potential impacts of catastrophic inheritance as a significant challenge in the era of LFMs. In the following, we outline the primary difficulties on addressing it effectively and efficiently.

**Availability Issue**. The foremost obstacle is the assessment of pre-trained LFMs $\mathcal{M}$ and their pre-training data $\mathcal{D}_{up}$. While some models like LLaMA2 (Touvron et al., 2023b) and Mistral (Jiang et al., 2023) are open-source, their performance usually fall behind proprietary counterparts such as GPT-4 (OpenAI, 2023) and Gemini (Google, 2023). A recent notable effort, LLM-360 (Liu et al., 2023b), strives to provide more comprehensive open-source training details. The proprietary nature of models and datasets creates a "black box" environment for users and researchers, limiting our ability to identify biases and analyze the impacts. Additionally, the massive scale of these models demand substantial computational resources, even when they are open-source, making detailed exploration more challenging.

**Evaluation Complexities**. Another challenge is the evaluation of intelligence in LFMs (Chang et al., 2023). The evaluation should not only be conducted w.r.t. models in standard benchmarks but also in society with diverse human-AI interactions. Traditional benchmarks are inadequate for these assessments. The strong performance of LFMs is challenged due to potential data contamination (Roberts et al., 2023; Schaeffer, 2023; Jiang et al., 2024), inappropriate metrics (Schaeffer et al., 2023; Sun et al., 2023), or lack of standards (Zhu et al., 2023b; Lei et al., 2023). Furthermore, latent biases in LFMs mean that many potential harms remain invisible until they manifest in real-world outcomes, making proactive evaluation and mitigation more difficult.

**Lack of understanding on LFMs**. Third, while there is great advance in understanding the generalization of modern neural networks, the specific influence of pre-training data biases on this aspect, i.e., the format of $g$, particularly in real-world scenarios, is less explored. Most of the case studies in Table 1 are limited to well-structured and small-scale datasets, which cannot thoroughly represent the complex real-world data LFMs may encounter. Thus, applying existing theories to LFMs, interpreting their real-world behavior, and assessing their societal impact are still formidable tasks.

**Trade-off in mitigation**. Finally, despite some early attempts (Chen et al., 2024b; Lu et al., 2023), addressing pre-training biases involves a delicate balance between efficiency and effectiveness. Re-training LFMs on bias-free data is ideal, but not always feasible. Black-box methods can mitigate specific biases but might overlook or intensify others in different contexts. This creates difficulties for researchers in optimizing and mitigating catastrophic inheritance in downstream tasks without sacrificing performance or inadvertently increasing biases in other aspects.

## 4 Our UIM Framework

In this section, we present the UIM framework, as depicted in Figure 2, to address catastrophic inheritance. UIM calls for future research from three perspectives: understanding catastrophic inheritance from pre-training and evaluation in downstream tasks to figure out the trend of function $g$ and $f$, interpreting the potential impacts and implications of biased pre-training data on downstream tasks both empirically and theoretically to find the form of $g$ and $f$, and mitigating the adverse effects of biased pre-training data on downstream tasks to reduce $g$ with the identified function relationship. It also serves as a general framework for studying CI, which we will show several existing works have already utlized it.

## 4.1 Understanding Catastrophic Inheritance

Fully understanding the impacts of catastrophic inheritance corresponds to finding the changes in both $f$ and $g$ from pre-training and downstream tasks respectively, including conducting empirical experiments and building novel evaluation metrics and benchmarks on various downstream tasks.

**Probing into Effects at Pre-training and Downstream**. The initial focus would be identifying the exact effects and the changes of them w.r.t. pre-training data biases at both the pre-training stage and downstream transferring stage. It is critical to study various types of pre-training data biases in this stage and find out the effects of such biases through large-scale experiments. As discussed earlier, the pre-training data biases not only shape the learning dynamics but consequently imprint on the model's behavior on downstream tasks (Nakkiran et al., 2019; Dar et al., 2021; Caballero et al., 2022). A particular aspect of interest is the relationship between these biases and scaling laws. We propose future research encompassing a comprehensive empirical investigation of different LFMs including CLIP (Radford et al., 2021), language models (Touvron et al., 2023a,b), and etc., under controlled and varying pre-training bias conditions. This investigation involves introducing different types and scales of synthetic and realistic biases into clean and controllable large-scale pre-training data, and illuminates the trend and changes in training dynamics, model behaviors, and generalization on downstream tasks. Changes of many other properties in $f$ and $g$ w.r.t biases is also worth studying, such as the expressive capacity (Zhang et al., 2016), transition from memorization to generalization (Power et al., 2022; Kumar et al., 2023; Davies et al., 2023), and structures of LFMs affected by the biases. Studying on the biases within controlled subset of concepts is also necessary (Feldman, 2020).

On the downstream side, we need to consider broader contexts to figure out the trend of $g$ to biases. This includes evaluating the LFMs on diverse downstream datasets $\mathcal{D}_{\mathrm{down}}$ of various domain and settings, such as tasks with imbalanced (Huang et al., 2016), noisy (Natarajan et al., 2013), few-shot (Wang et al., 2020), unlabeled (Wang et al., 2022), and ood (Hendrycks et al., 2021; Zhang et al., 2019) data, and different tuning algorithms $\mathcal{A}_{\mathrm{down}}$, such as prompt strategies (Zhu et al., 2023c,d), linear probing, parameter-efficient tuning (He et al., 2021), and even full tuning. A more comprehensive evaluation not only facilitates identifying the trend of $f$ and $g$ but also the compositional relationship of them.

**Evaluation Metrics/Benchmarks of LFMs**. LFMs are difficult to evaluate holistically, not only because of their complex capability but also the lack of proper evaluation metrics and benchmarks. We advocate for the development of new evaluation metrics that go beyond traditional performance measures on downstream tasks. The metrics should incorporate different aspects of LFMs, such as (adversarial) robustness (Wang et al., 2023a), fairness (Du et al., 2020; Nanda et al., 2021), bias (Wu and Aji, 2023), security, and privacy (Yao et al., 2023). Furthermore, evaluations must address the potential misalignment between LFM behaviors and ethical or societal norms, ensuring that these powerful models act in ways that are beneficial and non-harmful to society at scale. It is important to design metrics that measure the explicit influence and memorization of biased samples (Feldman and Zhang, 2020; Carlini et al., 2022). Establishing novel and robust evaluation benchmarks is also vital, considering the prevalence of data contamination that obscures the true capabilities of LFMs. Dynamic evaluation protocols (Zhu et al., 2023b; Fan et al., 2023b) represent a

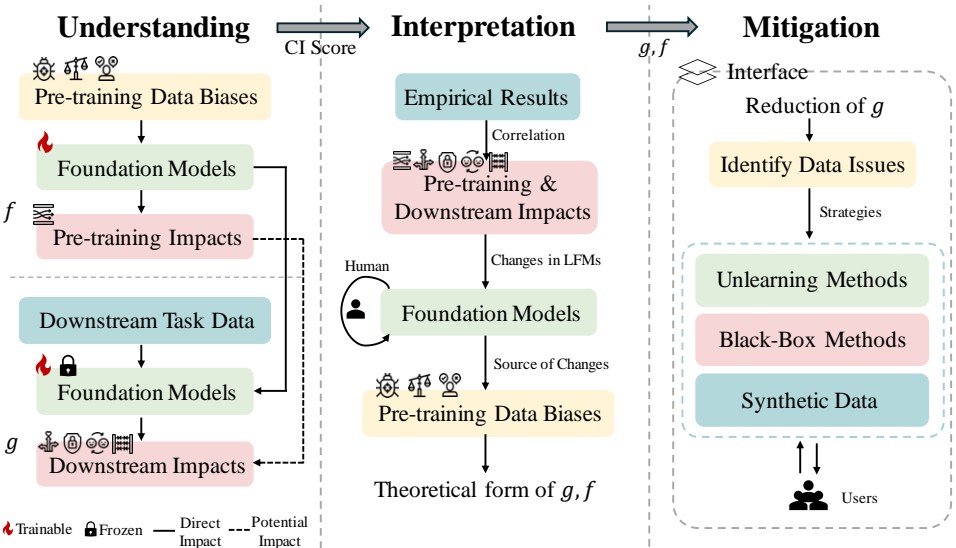

Figure 2: The UIM framework addressing catastrophic inheritance from understanding, interpretation, and mitigation.

promising direction. Future benchmarks should strive to generate rephrased, non-overlapping samples to counteract data contamination (Yang et al., 2023a). Additionally, collecting and annotating failure cases specifically arising from pre-training data biases at the deployment of LFMs will provide valuable insights for refining them. These dimensions are critical for understanding of catastrophic inheritance.

**Understanding the Societal Impact.** The assessment of catastrophic inheritance should be considered in an interdisciplinary fashion. It should broadly includes the human interaction with LLMs from the psychology aspects (Li et al., 2023b,a), and agents interaction within LLMs for studying society behaviors (Zhao et al., 2023b; Leng and Yuan, 2023; Park et al., 2023). We should also evaluate the effect of biases on critical applications of LFMs, such as medical and science (Singhal et al., 2022; Nejjar et al., 2023; Thirunavukarasu et al., 2023; Anderljung et al., 2023).

Several existing works have studied the understanding of LFMs from the perspective of pre-training data biases. For example, Zhu et al. (2023e) studied the data credibility issues in the pre-training of language models. Chen et al. (2024c,a) researched on the effects of pre-training corruption on CLIP and Diffusion models, respectively. Hu et al. (2024) studied the inheritance of adversarial examples from pre-training to downstream tasks of LFMs. Through a large-scale empirical study, Chen et al. (2024d) revealed the amplification effects of pre-training biases in diffusion models. All relevant works demonstrate that performing large-scale and well-controlled experiments about pre-training biases are essential to understanding their effects.

## 4.2 Interpreting the Impacts to Downstream Tasks

Interpreting why and how the malicious effects of pre-training data biases function in the downstream is crucial to address catastrophic inheritance, i.e., the exact form of $g$.

**Empirical Interpretation of Malicious Effect**. To empirically interpret the malicious effects of pre-training data biases, we need to conduct in-depth case studies and analyses on specific downstream applications. This involves examining the feature space using tools such as SVD, PCA, and T-SNE, both qualitatively and quantitatively. The singular values and vectors of the pre-trained feautres are often related to the transferability of generalization (Chen et al., 2024b; Xue et al., 2022; Chen et al., 2019). Jacobian matrix analysis is another perspective to explore transferability (Oymak et al., 2019), although its calculation in LFMs may require approximations (Yao et al., 2020). Such an empirical analysis also needs to be performed on a wide range of downstream tasks from different domains to assess $g$.

**Theoretical Interpretation of Catastrophic Inheritance**. In developing a theoretical interpretation of catastrophic inheritance in LFMs, we focus on frameworks that can precisely predict and articulate the observed bias inheritance from pre-training data to downstream tasks. This requires an in-depth examination of LFMs' internal mechanisms, particularly how they process and retain information from pre-training phases. Key to this exploration are concepts such as the balance between memorization and generalization (Zhang et al., 2016; Kumar et al., 2023; Zhu et al., 2024), the delineation of the memorization and generalization bounds (Kawaguchi et al., 2017), and also the theoretical evolution of LFM architectures during training. We aim to identify specific thresholds or critical points where pre-training biases critically influence these balances, similar in Nakkiran et al. (2019); Zhu et al. (2024). The theoretical frameworks help us to model the exact form of $g$.

**Interpretation based on Human-AI Collaboration**. The direction of adopting LFMs to correct themselves based on minimal and necessary human collaboration is also promising on interpreting the effects of pre-training biases (Jang, 2023). It involves design self-criticize (Wang et al., 2023b) and self-feedback loop based on human feedback to produce explanation and diagnosis (Gou et al., 2023) of LFMs themselves on the failures inherited from pre-training biases.

### 4.3 Mitigating Catastrophic Inheritance

Understanding and interpreting the malicious impacts on downstream tasks will help us design mitigation strategies.

**Black-Box Tuning Methods**. Black-box tuning is one of the most interesting methods for mitigating the malicious effects of the pre-training data biases on downstream tasks. These involve designing lightweight modules, such as additional layers, which can be applied to LFMs without altering their pre-trained weights. This approach is particularly intriguing due to its potential to remodel the feature space based on the biases identified in our empirical and theoretical analyses (Chen et al., 2019, 2024b). Similar methods have also been adopted in mitigating the adversarial noise of LFMs, especially in medical domain (Han et al., 2024). While parameter-efficient tuning methods share similarities with black-box approaches (Oh et al., 2023; Guo et al., 2023; Lin et al., 2023; Yu et al., 2023), they often require access to the internal structures or weights (He et al., 2021). Recently, Tong et al. (2024) have also tried tuning methods combining multiple models to mitigate the inheritance of a single model. Kim et al. (2024) proposed re-weighting methods along diffusion steps, specific to diffusion models, to learn unbiased diffusion models from biased datasets. Nonetheless, these black-box methods also present limitations, primarily due to the limited

scope of transformation they offer and the need to keep the pre-trained part of the model frozen. Future research will devise specialized regularization terms tailored to counteract the malicious effects of pre-training data biases, enhancing the effectiveness of these tuning methods.

**Unlearning Methods**. Machine unlearning techniques (Bourtoule et al., 2021; Xu et al., 2023a; Chen and Yang, 2023; Zhang et al., 2024a) can also be utilized to mitigate data biases prior to training in LFMs. The goal is to revise the LFM's knowledge to effectively forget, edit, and minimize the impact of biased data. Unlearning has also been adapted for diffusion models as demonstrated in (Wu et al., 2024), targeting the removal of learned biases. However, unlearned diffusion models have been found still generate unsafe contents recently (Zhang et al., 2023b). Chen and Yang (2023) introduced an efficient approach by integrating unlearning layers within transformer blocks to unlearn concepts in LFMs. Future developments should focus on designing novel methods requiring minimal interaction with the core structure and pre-training data of LFMs (Xiao et al., 2023) that effectively minimizes and unlearns the knowledge of certain types of biases from pre-training in LFMs. Addressing the trade-off between the bias mitigation at downstream and the performance degradation, as shown in existing works, is also an important question for future research.

**Synthetic Data Tuning Methods**. Synthetic data can also be utilized facilitate black-box tuning or unlearning methods in the situations where targeted biased data is inaccessible. The use of unbiased synthetic samples for further tuning, as demonstrated by Zhao et al. (2023a), can alleviate the effects of biases inherited from pre-training. Large diffusion models can be employed to generate the unbiased samples, which can be used to fine-tune LFMs (in black-box manners or for unlearning). Zheng et al. (2024) utilized synthetic data to solve the noisy label learning problem. Similarly, Seo et al. (2024) proposed to use synthetic c data for continual learning of LFMs. One promising research direction is to study to what extent the unbiased knowledge would be eliminated and how much unbiased data will be needed with synthetic data.

**Pre-traning Data Curation and Pruning**. Refining the pre-training dataset is a direct approach to mitigating biases, yet usually resource-intensive. Advanced tools for data inspection and documentation will be crucial in this process. Researchers will need to develop sophisticated metrics to effectively measure and balance the diversity, quality, and quantity of pre-training data, ensuring that the curated dataset is representative and free from harmful biases.

**Designing Lifelong Interfaces.** Novel platforms supporting lifelong updates of LFMs should be built, integrating the functions of identifying, understanding, interpreting, and mitigating the pre-training biases. After the stage of pre-training, the failures and misaligned behaviors of LFMs should be easily edited via the interaction of human on the platform to continually update the LFMs without re-training.

## 5 Conclusion

In this paper, we have identified an important yet neglected topic of LFMs, termed Catastrophic Inheritance, and delved into the multifaceted challenge of it, highlighting the critical need for understanding, interpreting, and mitigating the pre-training data biases. Our proposed UIM framework provides a comprehensive approach to understanding and ad-

dressing these issues. Through innovative methods such as black-box tuning, machine unlearning, synthetic data tuning, and pre-training data curation, we aim to advance the field in developing more robust, unbiased, and responsible LFMs. The future of LFMs depends on our ability to effectively manage and overcome the inherent biases in pre-training data. We hope this position paper inspires more research that contributes not only to the theoretical understanding of LFMs, but also to practical solutions to enhance their reliability and applicability in real-world scenarios.

## 6 Broader Impact

The research on Catastrophic Inheritance in LFMs addresses crucial concerns about the biases and limitations inherited from large-scale pre-training datasets. This work has significant implications across various domains. By highlighting the potential for these models to perpetuate and amplify biases, the study underscores the need for more responsible AI development and deployment. The proposed UIM framework aims to foster collaboration between machine learning and social sciences to better understand, interpret, and mitigate these biases. This interdisciplinary approach is essential for developing LFMs that are not only technically robust but also ethically sound and socially beneficial. We hope this position paper can guide future efforts in dataset curation, model training, and evaluation, ultimately contributing to the creation of fairer and more reliable AI systems.

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
