# OpenReview forum: "On Catastrophic Inheritance of Large Foundation Models"
_DMLR — Accepted by DMLR_

### Review · Reviewer_XpQS · 2024-06-09

**Recommendation:** 4
**Confidence:** 2

**Summary Of Contributions:**

In this position paper, authors discuss the issue of "Catastrophic Inheritance" in large foundation models (LFMs), which describes the negative effects and limitations that LFMs inherit from biased, large-scale pre-training data, which then adversely affect their performance on downstream tasks. Moreover, it is hard to interpret. The paper highlights various issues, such as biases, lack of generalization, and ethical concerns, which are transferred from pre-training data to LFMs. The topic is of great interest to the data-centric ML community. Finally, the UIM (Understand, Interpret, and Mitigate) framework is proposed to understand, interpret and mitigate the catastrophic inheritance of LFM.

**Strengths:**

see above comments

**Audience:**

Yes

**Broader Impact Concerns:**

I have no concerns about the broader impact of this paper. The authors provide detailed descriptions of their broader impact.

**Claims And Evidence:**

yes

**Datasets And Benchmarks:**

not applicable

**Extended Submissions:**

not applicable

**Limitations:**

see above weaknesses

**Requested Changes:**

see above weaknesses

**Strengths And Weaknesses:**

Strengths:

[0] Overall, this paper is well-organized, the figures are well-designed, and have positive impact to the community. It calls for more attention in this direction, which is crucial as LFMs are achieving amazing results in many applications and one of the core components is the data used for training. Identification and mitigation of the issues in data (such as bias) will benefit the healthy development of LFMs.

[1] The topic discussed is interesting and important. This manuscript identifies and highlights the often-overlooked problem of catastrophic inheritance in LFMs, bringing attention to a crucial area in LFM research.

[2] The proposed UIM framework provides a structured approach to understanding, interpreting, and mitigating the issues related to catastrophic inheritance.

[3] This paper provides a thorough review of existing studies, which provides context and supporting evidence about the impacts of pre-training data biases, etc.


Major weaknesses:

[1] This is not an actual weakness, is more like a discussion. One interesting topic is whether LFMs can outperform humans and extend the boundary of knowledge in human society. While it might not be the focus of this study, it will be appreciated if authors have some discussions about it. Currently, LFMs rely on labeled or unlabeled data generated by citizens or domain experts. A natural question is whether LFMs generate new knowledge or advance the science of our society, since their knowledge comes from the data we fed them. While we are witnessing LFMs outperform experts on some tasks, will it extend to all settings?

[2] This study mainly discusses language, vision, and multi-modality models. Another direction, time-series foundation models, are receiving growing attention recently. Will the findings from this study generalize well to time-series LFMs?

[3] Data are generated by human from diverse background, e.g., culture, education, and country. Sometimes people have very different views towards the same thing and different countries have different rules or laws. For example, about euthanasia, abortion, etc. While data from different countries about these topics are different, it is hard to say which answer is correct. Therefore, how to address such “bias” is very important. Maybe the authors can have more discussions and it will benefit future studies in this area.

---

### Review · Reviewer_zHir · 2024-07-02

**Recommendation:** 3
**Confidence:** 2

**Summary Of Contributions:**

This work introduces the framework “UIM” to understand, interpret, and mitigate “catastrophic inheritance” of harmful training data patterns in foundation models. It includes an extensive literature review related to issues with training data in foundation model and establishes a research direction towards a comprehensive approach of understanding and mitigating these issues.

**Strengths:**

The work contains notable strengths including an extensive literature review and comprehensive framework for understanding, interpreting, and mitigating training data issues in large foundation models. Please see Strengths listed above.

**Audience:**

Yes

**Claims And Evidence:**

Yes

**Datasets And Benchmarks:**

N/A

**Extended Submissions:**

N/A

**Limitations:**

Please see Weaknesses above.

**Requested Changes:**

Discussed in the Weaknesses above, the paper would be strengthened by clarifying how the UIM framework enables improved research and better defining catastrophic inheritance such that it reconciles the many definitions of catastrophic and malicious impacts or clarifying its main contribution in literature review.

**Strengths And Weaknesses:**

Strengths
* The literature review is quite extensive, covering different kinds of training data issues such as skewed distribution, unethical content, and inspecting tools, as well as how they propagate to downstream tasks via a lens of generalization and privacy/security.
* The paper contains concise summary tables with examples of known training data biases / catastrophic inheritance, making it easier to reference clear examples of these harms
* Via the UIM framework, this paper usefully defines an area of work that guides a thorough analysis of how harmful training data issues can impact large foundation models. This includes a variety of research directions, spanning improved probes and benchmarking efforts, methods of understanding propagation to downstream tasks, and mitigating catastrophic inheritance
* The contribution is well-motivated by the important and relevant area of responsible foundation model development, and is well aligned with DMLR.
* The paper is well-written and clear.

Weaknesses
* While the work does a great job at outlining the large and complex problem of catastrophic inheritance, its main contributions are via literature review and framework outlining existing and previous research directions, rather than including new analyses, findings, or contributions related to the topic. Furthermore, it's not clear how the UIM framework allows for materially improved research at this point.
* The definition of "catastrophic inheritance" does not well capture what is meant by "catastrophic" and "malicious" impacts of large foundation models. The authors seem to use it to cover a broad set of well-known problems with large foundation models but it's not clear to what extent the framing of "catastrophic inheritance" provides new insights into existing issues with foundation models.
* Compared to the rest of Section 2, Section 2.4 Mitigations feels a bit spare and would benefit from additional specificity.

---

### Review · Reviewer_gAAF · 2024-07-14

**Recommendation:** 3
**Confidence:** 2

**Summary Of Contributions:**

Authors introduce the concept of Catastrophic Inheritance (CI), which they define as the negative impacts of adapting large foundation models (LFMs) to downstream tasks due to biases present in the large-scale pre-training data. The authors provide a comprehensive review of existing literature related to CI, discussing various types of biases in pre-training data, their potential impacts on downstream tasks, and existing mitigation strategies. They propose a research framework called UIM (Understand, Interpret, Mitigate) to address CI. The UIM framework aims to:

1. Understand the effects of biases on pre-training dynamics, generalization behaviors, scaling laws, and downstream tasks through empirical experiments and the development of new evaluation metrics and benchmarks.
2. Interpret the fundamental reasons for the effects of biased pre-training data on downstream tasks, both empirically and theoretically.
3. Mitigate the adverse effects of biased pre-training data on downstream tasks using black-box tuning methods, unlearning techniques, synthetic data tuning methods, and pre-training data curation.

**Strengths:**

1. Provides a clear and concise definition of Catastrophic Inheritance (CI), a novel concept in the field of large foundation models (LFMs).
2. Offers a comprehensive overview of the different types of biases present in pre-training data and their potential negative impacts on downstream tasks; hence may act as a decent review paper.
3. Proposes a structured research framework (UIM) to guide future research in understanding, interpreting, and mitigating CI.

**Audience:**

No

**Broader Impact Concerns:**

I do not have any broader impact concerns.

**Claims And Evidence:**

Sufficient related work is cited. Since the paper does not propose a new approach, there are no claims to be checked.

**Datasets And Benchmarks:**

This is not a dataset or a benchmark.

**Extended Submissions:**

I believe this is not an extended submission.

**Limitations:**

1. Lacks novel technical contributions or empirical analyses to support the proposed UIM framework.
2. Does not offer concrete recommendations or baselines for interpreting and mitigating the effects of CI in practical applications.
3. Fails to address the potential trade-offs and challenges associated with different mitigation strategies, such as the balance between bias reduction and model performance.

**Requested Changes:**

1. The authors could provide more detailed explanations and examples of how the UIM framework can be applied in practice -- including specific methodologies for understanding, interpreting, and mitigating catastrophic inheritance, as well as case studies demonstrating the effectiveness of the framework.
2. The authors could conduct experiments to quantify the impact of different types of biases on downstream tasks and evaluate the effectiveness of various mitigation strategies with simple baselines.
3. While the paper discusses several mitigation strategies, it could benefit from a more in-depth exploration of these approaches -- including a small comparative analysis of different methods, a discussion of their limitations, and suggestions for future research directions.

Currently, this reads as a very broad review paper.

Post rebuttal, I raised my score to Leaning to Accept.

**Strengths And Weaknesses:**

Strengths:

1. Provides a clear and concise definition of Catastrophic Inheritance (CI), a novel concept to the best of my knowledge in the field of large foundation models (LFMs).
2. Offers a comprehensive overview of the different types of biases present in pre-training data and their potential negative impacts on downstream tasks; hence may act as a decent review paper.
3. Proposes a structured research framework (UIM) to guide future research in understanding, interpreting, and mitigating CI.


Weaknesses:

1. Lacks novel technical contributions or empirical analyses to support the proposed UIM framework.
2. Does not offer concrete recommendations or baselines for interpreting and mitigating the effects of CI in practical applications.
3. Fails to address the potential trade-offs and challenges associated with different mitigation strategies, such as the balance between bias reduction and model performance.